# A new tandem repeat-based genotyping scheme for the global surveillance of *Xanthomonas citri* pv. *mangiferaeindica*e, an understudied bacterial pathogen of major importance to mango and cashew production

Karine Boyer[1], Cyrille Zombre[2], Laure Payan[1], Issa Wonni[2], Olivier Pruvost [1]*

**1** CIRAD, UMR PVBMT, Saint Pierre, La Réunion, France, **2** INERA, Station de Farako-Bâ, Bobo-Dioulasso, Burkina Faso

* olivier.pruvost@cirad.fr

## Abstract

Bacterial canker, caused by *Xanthomonas citri* pv. *mangiferaeindicae* (*Xcm*), is a disease that has a devastating impact on mango and cashew industries in many regions. Yet, despite its agricultural importance for these Anacardiaceae species, *Xcm* has been neglected. Little is known about its epidemiology, evolution and molecular interactions with host plants. The most relevant studies reporting its genetic structure were primarily based on amplified fragment length polymorphism (AFLP) data. This technique provides reliable assessments of the genetic relatedness among bacteria, but is limited in terms of interlaboratory comparisons. Alternative genotyping techniques are required to decipher the global epidemiology and geographic expansion of *Xcm*. Herein, we screened the genome of the *Xcm* strain CFBP1716 for tandem repeats. We developed and evaluated the performance of an optimized Multi Locus Variable number of tandem repeat Analysis (MLVA), targeting 16 tandem repeat loci primarily with large repeat units, *i.e.,* minisatellites (MLVA-16). To achieve this, we genotyped a comprehensive collection of 152 *Xcm* strains, representative of the pathogen's worldwide genetic diversity, together with some reference strains of *X. citri* pv. *anacardii*, another genetically-related pathogen of Anacardiaceae. MLVA-16 allowed us to distinguish the two pathovars. Although MLVA-16 was slightly less discriminative than AFLP, the two derived datasets were strongly correlated, suggesting that MLVA-16 provides a good phylogenetic signal. Five clusters with some geographic coherence were delineated, based on discriminant analysis of principal components. The two major clusters grouped strains from multiple geographic origins. In contrast, all strains that have emerged on mango or cashew in West Africa grouped in one cluster, which did not contain any strains of different origin. MLVA-16 represents an opportunity to improve our understanding of the structure of *Xcm*

**Data availability statement:** All data are available from the MLVAbank database https://webphim.ird.fr/MLVA_Bank/Genotyping/index.php?&largeur=1920.

**Funding:** KB and OP report the following sources of funding: The European Union (ERDF contracts GURDT I2016-1731-0006632 and 2024-1248-005756) https://commission.europa.eu/funding-tenders/find-funding/eu-funding-programmes/european-regional-development-fund-erdf_en Conseil Régional de La Réunion https://regionreunion.com Centre de coopération internationale en recherche agronomique pour le développement (CIRAD) https://www.cirad.fr.

**Competing interests:** The authors have declared that no competing interests exist.

populations, by sharing genotyping data. The MLVA-16 data generated in this study was deposited in a dedicated online database.

## Introduction

Protecting crops from pests and diseases is a priority for global food security. Pathogens not only cause significant economic losses, they also directly affect food availability, causing food insecurity [1]. Surveillance programs are essential to help mitigate these potentially serious social and economic consequences. They can provide important data on the early detection of outbreaks and guidance for disease management. Genotyping techniques targeting DNA markers or single nucleotide polymorphisms (SNPs), identified from whole genome sequencing (WGS) data, can be applied to the study of outbreak pathogen populations. Indeed, they can determine important factors to help improve disease management, such as the source populations, transmission pathways and migration distances. The epidemiology of bacterial pathogens has greatly benefited from high throuput WGS [2,3] over the last decade. However, there are still many cases where marker-based genotyping is useful. Among markers with desirable characteristics, tandem repeats (TR), assayed in a Multi Locus Variable Number of Tandem Repeat (VNTR) Analysis (MLVA) format, are simple, discriminative, robust and cheap. They can be used to (i) decipher the population biology of bacterial pathogens, especially in the case of the so-called genetically monomorphic pathogens, or (ii) provide guidance for strain selection in studies on WGS-based population genomics [4,5]. Selecting TR markers with a rate of evolution that fits the evolutionary scale under investigation is key [6]. For studies at a large evolutionary scale, MLVA typing schemes that target a large number of minisatellites (defined here as TRs with a repeat unit length ranging from 10 to 300 bp), can produce datasets with a strong phylogenetic signal [7].

Two Anacardiaceae species, mango (*Mangifera indica* L.) and cashew (*Anacardium occidentale* L.) represent major cash crops in tropical and subtropical countries. Mango world production reached approximatively 61 M tons in 2023 and ranked 6th in fruit production (https://www.fao.org/faostat/en/#data/QCL). The largest mango producers, India, Indonesia, China, Mexico and Pakistan, represent 65% of global production. Cashew world production reached approximatively 4 M tons in 2023, of which 43% came from the top 5 producers in Africa (Ivory Coast, Benin and Tanzania) and Asia (India and Vietnam). With the exception of Mexico and Vietnam, production in the major mango- and cashew-producing countries is threatened by the potentially destructive mango bacterial canker (MBC), also called mango bacterial black spot (MBS) [8–10]. The earliest reports of MBC were in India (herbarium specimens collected in Bihar) in 1881 [11] and South Africa (outbreak description) in 1915 [12]. MBC occurs in many countries in Asia, Oceania, Africa, the Indian Ocean region and North America [8,13,14].

The causal agent of MBC was first described in the early 20th century and classified in the 1970s as *Xanthomonas campestris* pv. *mangiferaeindicae* [15]. Then,

with the evolution of taxonomy, it was reclassified as *Xanthomonas citri* pv. *mangiferaeindicae* (*Xcm*) [16,17]. *Xcm* is a Gram-negative bacterium that produces non-pigmented colonies, unlike most *Xanthomonas* species [8,18]. *Xcm* was first isolated from the mango tree, but was found to be pathogenic to other Anacardiaceae species by inoculation [11,19]. *Xcm* strains were subsequently shown to induce symptoms under natural conditions on other members of the *Anacardiaceae* family, including cashew, Brazilian pepper [8,14,20]. The genetic diversity among *Xanthomonas* strains isolated from Anacardiaceae worldwide was previously studied using several genotyping techniques, for example, using amplified fragment length polymorphism (AFLP) and a derivative of AFLP targeting an insertion sequence (IS-LMPCR). Restriction fragment length polymorphism (RFLP) has also been used to target different genomic regions, *i.e.*, IS*1595*, the *hrp* cluster and a transcription activator-like type III effector (TALE) [16,18,21]. AFLP combines a high discriminatory power with a good phylogenetic signal and was most suited to deciphering the global *Xcm* genetic structure [16,22]. Among strains isolated from *Anacardiaceae*, other than mango, some strains were confirmed as *Xcm*, while others were reclassified into distinct pathovars, such as *X. citri* pv. *anacardii* (*Xca*) and *X. axonopodis* pv. *spondiae* [14,22]. These studies also suggested that the *Xcm* population structure has three genetic clusters (referred to as A, B and CD to match the early RFLP-based clustering). Among these genotyping techniques, only TALE-RFLP delineated strains of cluster CD, based on their host of origin (mango vs. Brazilian pepper) [18,22]. However, these techniques are clearly limited in terms of their capacity to analyse large strain numbers (RFLP) or to compare datasets produced in different laboratories (AFLP). Therefore, new typing techniques are required [23].

Clustered regularly interspaced short palindromic repeats (CRISPR) and tandem repeats (TR) were recently proposed as useful targets for deciphering the genetic diversity of some xanthomonads [24]. CRISPR loci are only present in some *Xanthomonas* pathovars [23,25,26]. In contrast, TRs, referred to as microsatellites and minisatellites according to the size of the repeats, are widely present in *Xanthomonas* genomes. Microsatellites (TRs < 10 bp in size) are primarily used for delineating the structure of populations at small evolutionary scales (*i.e.*, local molecular epidemiology). However, high mutation rates make them suboptimal for assessing deep genetic relatedness among populations. Minisatellites (TRs ranging from 10 to 250 bp in size) display a less rapid evolution, compared to microsatellites. Minisatellite typing, which was previously developed for the genetically related *X. citri* pv. *citri,* showed a phylogenetic structure congruently matching that derived from AFLP and single nucleotide polymorphism (SNP) data [27,28]. Similarly, minisatellites were recently found to be useful for assessing the global genetic diversity of the cassava pathogen, *X. phaseoli* pv. *manihotis* [29]. Given the technical simplicity and portability of minisatellite typing, *i.e.*, good interlaboratory reproducibility, availability of online access databases, such as MLVAbank [30], this technique is suitable for describing the overall structure of pathogen populations (*i.e.*, global molecular epidemiology) [28,31]. It is superior to older typing methods, such as RFLP and AFLP.

The aims of this study were to (i) evaluate CRISPR and TR suitability for subtyping *Xcm*, and (ii) develop a robust MLVA scheme suitable for global studies on its epidemiology and to build a dedicated online public database. Herein, we report a new MLVA genotyping scheme targeting 14 minisatellites and two microsatellites (MLVA-16). We demonstrate its pertinence for studying the global diversity of *Xcm*.

## Materials and methods

### Bacterial cultures and media

In this study, we used 152 *Xcm* strains, originating from several continents and host species, and representative of its genetic and pathological diversity [14,22] (S1 Table). Solid cultures were performed on YPGA plates (yeast extract 7 g l$^{-1}$, peptone 7 g l$^{-1}$, glucose 7 g l$^{-1}$ and agar 18 g l$^{-1}$; propiconazole 20 mg l$^{-1}$; pH 7.2), at 28°C. For each strain, a single colony was transferred on YPGA and grown at 28°C overnight. One µl of culture was diluted in 400 µl of 0.01M Tris buffer, pH 7.2, and stored at −30°C in deepwell microplates prior to PCR amplifications.

### *In silico* VNTR and CRISPR mining from genomic sequences of *Xcm*

The CFBP1716 *Xcm* genome sequence (Genbank accession CP156913), herein considered our reference, together with two other high-quality genome sequences produced by our group were used to detect TR loci using Tandem Repeat Finder (TRF) [32,33]. The total length was set in a range of 50–1000 bp, the length of tandem repeats was set at ≥10 bp, and other parameters were attributed default settings. Loci with TR sequence conservation < 80% and those corresponding to TALE genes were not considered further. Genomic flanking regions (500 bp each side of the TR region) were also used to define primer pairs for PCR amplification, with the Oligo 6 software (https://www.oligo.net/). CRISPR mining was performed on the available high quality *Xcm* genomes [25,26] using CRISPRCasFinder with default parameter settings [34]. The high-quality genome sequences of strains CFBP9184, CFBP9185 and GXG07 (Genbank accessions CP156909, CP156905 and CP073209, respectively) were screened *a posteriori* upon availability [25,26].

### MLVA genotyping

Genotyping was performed using a comprehensive strain collection representing the currently known genetic diversity worldwide [16] and three reference strains of *Xanthomonas citri* pv. *anacardii*, another *X. citri* pathovar that is pathogenic to Anacardiaceae (S1 Table). Primer pairs targeting single-locus alleles were used in a multiplex PCR format, using the Clonetech Terra PCR Direct Polymerase Mix (Takara Bio, CA, USA). For each primer pair, the reverse primer in the PCR mix was 5'-labeled with one of the following fluorescent dyes: 6-FAM, Yakima Yellow, ATTO 565 or Dragon Fly Orange (Eurogentec, Liege, Belgium). PCR mixes contained 7.5 µl of 2x Terra Buffer (containing a hot-start Taq DNA polymerase), 1.5 µl of 5x Q-Solution (Qiagen, Courtaboeuf, France), 0.2 to 0.6 µM of each primer (S2 Table), 1 µl of purified culture and RNase-free water to yield a final volume of 15 µl. PCR amplifications were performed in a Veriti thermocycler (Applied Biosystems, Courtaboeuf, France), under the following conditions: 2 min at 98°C for hot-start activation; followed by 25 cycles of denaturation at 98°C for 10 s, annealing at a temperature ranging from 62 to 69°C, extension at 68°C for 1 min or 1 min 30 s (for long amplicons), and a final extension step at 68°C for 30 min. We mixed 1 µl of diluted amplicons (diluted at a rate determined by test runs, with dilutions ranging from 1:40–1:120) with 10.6 µl of Hi-Di formamide and 0.4 µl of a GeneScan 600 LIZ V2, as an internal size standard (Applied Biosystems, Courtaboeuf, France), or 10.5 µl of Hi-Di formamide (Applied Biosystems, Courtaboeuf, France) and 0.5 µl of a GeneScan 1200 LIZ (Applied Biosystems, Courtaboeuf, France), depending on the size of amplicons. Then, the mixture was denatured for 5 min at 95°C and placed on ice for at least 5 min. Capillary electrophoresis was performed in an ABI PRISM 3500xl genetic analyzer (Applied Biosystems, Courtaboeuf, France), using a performance-optimized polymer, POP-7, at 15000V at 60°C, with an initial injection of 23 s. Reference sample CFBP1716 was used as an internal control in all experiments. Fragment sizes were determined using Genemapper V6.0 (Applied Biosystems, Courtaboeuf, France). In a few cases, no amplification was obtained by multiplex PCR. Corresponding samples were assayed again using the same primers in a simplex PCR format. If a lack of amplification was confirmed, more external primers were designed and tested. In this case, we used a long-template polymerase. Simplex PCRs were performed using the BD advantage 2 polymerase mix (Takara Bio, CA, USA). Briefly, 2 µl of boiled bacterial suspension was used as a template in mixes containing 5 µl of 10x Advantage 2 PCR Buffer SA 10x, 0.2 mM of dNTPs mix, 0.2 µM of each primer (S2 Table), 1 µl of Advantage 2 polymerase mix and RNase-free water to yield a final volume of 50 µl. PCR were performed in a Veriti thermocycler (Applied Biosystems, Courtaboeuf, France), under the following conditions: 3 min at 95°C followed by 40 cycles of denaturation at 95°C for 15 s, annealing at temperatures ranging from 62 to 69°C, extension at 68°C for 1 min and a final extension step at 68°C for 7 min. All simplex amplicons were visualized using a Qiaxcel advanced system (Qiagen, Courtaboeuf, France).

### TR sequencing

Rare alleles (n < 5) were checked by Sanger sequencing of amplicons. Amplicons were obtained by simplex PCR, with the same primers that were used for genotyping using the BD advantage 2 polymerase mix (Takara Bio, CA, USA), as

described before. Amplicons were then purified using Sera-Mag magnetic beads (Cytiva, Villacoublay, France), as recommended by the manufacturer. Once purified, amplicons were sequenced using the BigDye™ Terminator v.3.1 Cycle Sequencing Kit (Applied Biosystems, Courtaboeuf, France): 5–20 ng of PCR product were used in mixtures containing 2 μl of BigDye Terminator V3.1, 1 μl of 5x sequencing buffer, 1 μl of primer at 5 μM and water to yield a final volume of 10 μl. Sequencing reactions were performed in a Veriti thermal cycler according to the manufacturer's instructions. Sequencing products were purified using the BigDye XTerminator (Applied Biosystems, Courtaboeuf, France), as recommended by the manufacturer. Capillary electrophoresis was performed in an ABI PRISM 3500XL genetic analyzer (Applied Biosystems, Courtaboeuf, France) with a 50 cm capillary array.

### Data scoring and exploration

In a single case (Xcm-4431), TRF predicted different TR unit size alternatives. Genotyping data meant that the correct size could be deciphered unambiguously. Amplicon sizes were turned into tandem repeat numbers using Genemapper V6.0. Repeat numbers of TR arrays with truncated repeats were rounded up to the nearest higher integer. Nei's diversity index was calculated for each TR locus using poppr v.2.9.3 package in R. A minimum-spanning tree was built using PHYLOVIZ 2.0 [35,36], using an algorithm, which combines global optimal eBURST (goeBURST) and Euclidian distance best suited for MLVA data. The population structure of our strain collection was analyzed using Discriminant Analysis of Principal Components (DAPC), a multivariate method used to identify and describe clusters of genetically related strains using the adegenet v.2.1.7. R package [37].

MLVA data were compared to the reference method for *Xcm* typing, AFLP [16,22]. Our subset comprised 98 strains, for which genotyping data is available for both typing techniques. We compared the methods' discriminatory power by computing the Hunter Gaston discriminatory index (HGDI), using the DescTools package v.0.99.48 in R [38]. The correlation between distance matrices (Dice dissimilarity for AFLP and Manhattan's distance for MLVA, respectively) was tested by computing the Kendall's coefficient of concordance (W) among the distance matrices, using a permutation test (9,999 permutations) with the "CADM.global" function of the ape package v. 5.6.2 in R [39]. The non-parametric Kendall's W statistic evaluates congruence among multiple rating systems, with W ranging from 0 (no agreement) to 1 (complete agreement) [40].

## Results

### *In silico* analyses and TR marker selection

In this study, we identified minisatellites from the genome of the *Xcm* pathotype strain (CFBP1716), and then subsequently assessed their conservation level in other available high-quality genome sequences (Fig 1). A genotyping method was developed for selected markers, based on multiplex PCR and resolution of the fragments produced by capillary electrophoresis. Finally, a public database including all typing data was created, with the aim that the scientific community could use it for further studies.

CRISPRCasFinder suggested the absence of *cas* genes in *Xcm* genomes. We detected an array composed of five direct repeats and five spacers, which were monomorphic among all queried genomes. This indicates the inappropriateness of CRISPR elements for subtyping *Xcm* (S3 Table). TRF identified 23 TR loci that were evaluated using the *Xcm* strain collection and three reference *Xca* strains (S1 Table). No amplification was produced for *Xca* strains for 10 TR loci (Xcm0497, Xcm0794, Xcm0909, Xcm0943, Xcm2532, Xcm3117, Xcm3948, Xcm4232, Xcm4578 and Xcm4970) (Table 1). Moreover, three markers made it possible to distinguish *Xcm* and *Xca* strains. Xcm4486 was found monomorphic among *Xcm* strains and produced a different sized amplicon among *Xca* strains. Xcm2555 and Xcm3141 were found polymorphic among *Xcm*, but *Xca* strains yielded a different sized amplicon. Fourteen TR markers, therefore, have a diagnostic value at the pathovar level.

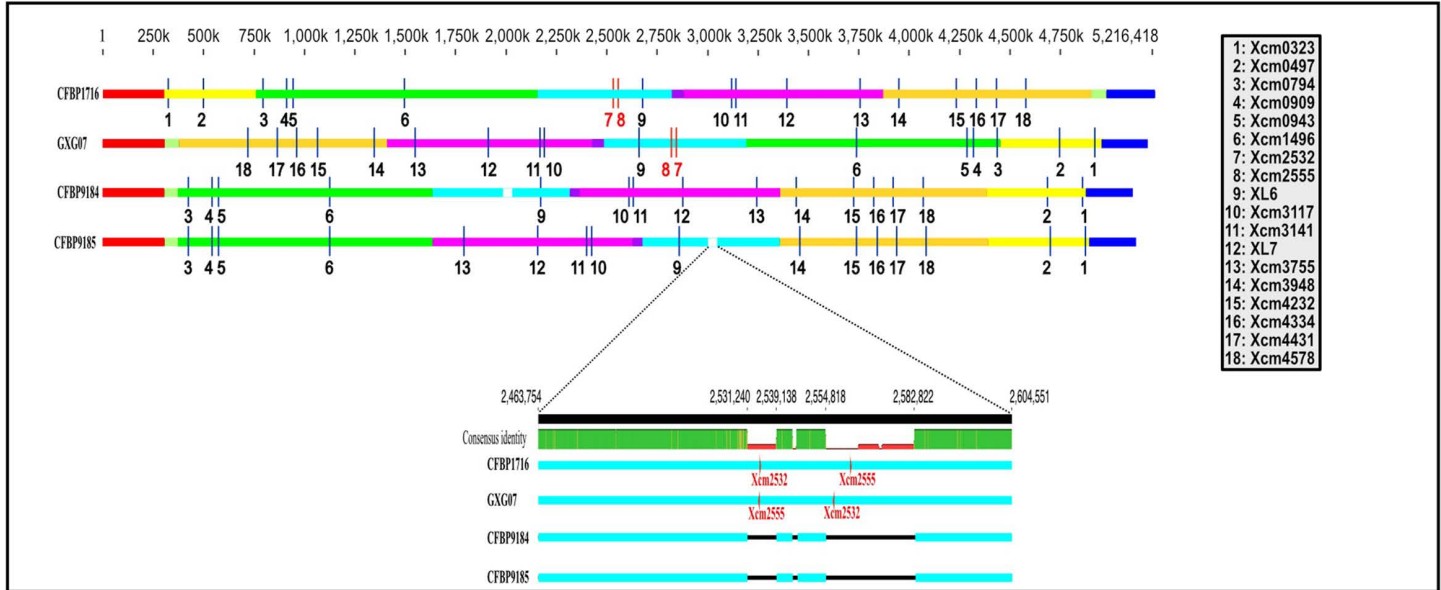

**Fig 1. Genomic location of TR markers used in MLVA-16, using *Xcm* pathotype strain (CFBP1716) as the reference.** Block colors refer to the MAUVE alignment performed in Geneious R10.2.6.

Six candidate TRs (Xcm2977, Xcm3763, Xcm3942, Xcm4486, Xcm4970 and Xcm5175) were found monomorphic for all *Xcm* strains assayed and were not considered further. No amplicon was produced from any West African strain for Xcm2532 and Xcm2555 in contrast with the *Xcm* strains of different origin. The concomitant absence of amplification at loci Xcm2532 and Xcm2555 has some epidemiological value, by allowing us to distinguish these unique strains (*i.e.*, the only ones reported to date causing severe outbreaks on cashew, in addition to mango). However, genotyping data derived from these two markers were not used further in the genetic diversity analysis.

The analysis of two high quality genome sequences of strains from Burkina Faso revealed that these two markers are part of two deletions (23 and 8 kb in size, respectively – Fig 1), which were detected within an integrative and conjugative element (ICE), as compared to CFBP1716 [26]. No amplicon was detected for strain JF955 from the Comoros for Xcm2532 (but one was obtained for Xcm2555). Conventional PCR yielded a 1.5 kb-long amplicon for this strain (therefore, it was undetectable by capillary electrophoresis). This amplicon was sequenced and showed the insertion of a 1,203 bp-long IS*Xcd1* (IS*407*-like element of the IS*3* family – accession AF263433), with 92% nucleotide identity over 100% length. The produced sequence allowed us to determine the number of repeats for strain JF955.

Fourteen minisatellite markers showed polymorphism and produced amplicons for all assayed Xcm samples. In order to increase its resolutive power, the derived dataset was supplemented with genotyping data previously obtained for two microsatellites displaying a low polymorphism rate, XL6 and XL7 [21].

## MLVA genotyping

All strains of *Xcm* could be genotyped, with TR units per locus ranging from 0 (no amplicon detected for Xcm2532 and Xcm2555) to 12 (Xcm0943) (Table 1). Rare alleles were identified for Xcm0909, Xcm1496, Xcm2532, Xcm37117, Xcm3141, Xcm3755 and Xcm4578. For these samples, Sanger sequencing confirmed that the typing results were indeed correct. We identified 46 haplotypes among the assayed strains and available high-quality genomes. The genotyping data and analysis of TRs in the four available high-quality *Xcm* genomes yielded identical results. MLVA-16 was slightly less

**Table 1. TR markers used in the MLVA-16 scheme on a worldwide strain collection of *Xanthomonas citri* pv. *mangiferaeindicae* and reference strains of *X. citri* pv. *anacardii*.**

| Locus | TR length (bp) | Position in CFBP1716 | Amplification in *Xca* | ORF putative function | Primer sequences (5' --> 3') | Range of repeat number | Number of alleles in *Xcm* (H$_T$) |
|---|---|---|---|---|---|---|---|
| Xcm0323 | 27 | 323 106 --> 323 186 | Yes | Intergenic | CCCGAGCCAAAC-CGAATCAC | 3-4 | 2 (0.389) |
| | | | | | YY-GCAGCCGACCC GCGCATCCA | | |
| Xcm0497 | 16 | 497 019 --> 497 165 | No | Intergenic | GCCTGCGTCGTCGAT-GACAT | 3-10 | 8 (0.795) |
| | | | | | ATTO565-CCTCCAAC GCGCAATACCGA | | |
| Xcm0794 | 15 | 794 526 --> 794 565 | No | Protein of unknown function | CCGAGTTCGCCG ACACTGCT | 0-4 | 3 (0.333) |
| | | | | | 6-FAM-AGTTTCTT CCACCGCTTCGTCCT | | |
| Xcm0909 | 22 | 909 920 --> 909 972 | No | Intergenic | ATCGCCACCCTGGA ACATGACA | 2-3 | 2 (0.051) |
| | | | | | YY-ATGTGGACAAG CCGCGCAAT | | |
| Xcm0943 | 33 | 943 695 --> 944 023 | No | Protein of unknown function | GTATGTTTGAA GGCTTCGAA | 5-12 | 7 (0.493) |
| | | | | | DFO-GGTTCATCTAAT CGGTTCGG | | |
| Xcm1496 | 11 | 1 496 267 --> 1 496 306 | Yes | Intergenic | GGCCACGGCTG CGAAACTCA | 3-4 | 3 (0.496) |
| | | | | | 6-FAM-GTGCCAACG CCACCGACGAT | | |
| Xcm2532 | 22 | 2 532 365 --> 2 532 427 | No | Replication protein C | ACCAAGACCAGCAA GCGATG | 0-4 | 2 (ND) |
| | | | | | ATTO565-CCGACGAA CCCGAATACCG | | |
| Xcm2555 | 23 | 2 555 950 --> 2 556 019 | Yes (d) | Intergenic | CGCCGCCTTTCT GGTTGAGA | 0-3 | 2 (ND) |
| | | | | | 6-FAM-GGCTGGAAC TGCTGACCTT | | |
| Xcm3117 | 11 | 3 117 938 --> 3 117 964 | No | Intergenic | GCTGCGGATAC AACTCTCGAA | 3-4 | 2 (0.013) |
| | | | | | ATTO565-CGATGCA GGTTTAGACATCCCA | | |
| Xcm3141 | 117 | 3 141 502 --> 3 141 852 | Yes | Intergenic | TGGAGTTGCG GCAGTCTTGA | 2-3 | 2 (0.039) |
| | | | | | 6-FAM-CGGTGGAGC GGTGGGTTA | | |
| Xcm3755 | 10 | 3 755 447 --> 3 755 473 | Yes | Intergenic | GCCGGTGACGA TGACTGTATCCA | 2-3 | 2 (0.013) |
| | | | | | DFO-AGCAGATGA GCAGCATTCCT | | |
| Xcm3948 | 166 | 3 948 716 --> 3 949 059 | No | Intergenic | TCGGCGATTATG CGTTCTGG | 2-3 | 2 (0.494) |
| | | | | | ATTO565-TTGCGGCT GGCTGTCGTTTG | | |

*(Continued)*

**Table 1.** (Continued)

| Locus | TR length (bp) | Position in CFBP1716 | Amplifica-tion in *Xca* | ORF putative function | Primer sequences (5' --> 3') | Range of repeat number | Number of alleles in *Xcm* (H$_T$) |
|---|---|---|---|---|---|---|---|
| Xcm4232 | 11 | 4 232 451 --> 4 232 488 | No | Intergenic | ATTGCTGCAGTT CCGTCCT | 3-4 | 2 (0.494) |
| | | | | | 6-FAM-TCGACCTCTT GCGGTTTCCAG | | |
| Xcm4334 | 172 | 4 334 965 --> 4 335 481 | Yes | Protein of unknown function | ACGACAGAACCC GGCTTATC | 2-3 | 2 (0.209) |
| | | | | | YY-CAGGCGGT GGAAGGGAGT | | |
| Xcm4431 | 42 | 4 431 522 --> 4 431 587 | Yes | Protein of unknown function | ACGCCACCAATTCT TGACGACT | 2-3 | 2 (0.389) |
| | | | | | YY-TGGAAAAA CCGCTCGGGCAAT | | |
| Xcm4578 | 133 | 4 578 967 --> 4 579 310 | No | Intergenic | ACACCATGGGC GCAGTCAAC | 2-5 | 3 (0.299) |
| | | | | | DFO-TGCCGCAG GGAATGGACCGA | | |
| XL6 [a] | 7 | 2 678 065 --> 2 678 121 | No | Intergenic | ATCGCACAGCA GCAACGAAGG | 3-8 | 6 (0.482) |
| | | | | | NED-TGACAAGCA GGAGCAGGCGCATGG | | |
| XL7 [a] | 7 | 3 392 261 --> 3 392 309 | No | Intergenic | TCCTGCGATGG CGAGTGG | 5-8 | 4 (0.397) |
| | | | | | PET-TGCGCAAGCT GGTCAAGTGG | | |
| | | | | | | **Mean** | **0.300** |

[a]from [21].

DFO: Dragon Fly Orange.

YY: Yakima Yellow.

discriminative than AFLP with Hunter's D values of 0.948 and 0.911 for AFLP and MLVA-16, respectively. Distance matrices derived from MLVA-16 and AFLP were highly correlated (K = 0.882; p = 0.0001). The mean Nei's genetic diversity (H$_T$) was 0.300 overall and ranged from 0.013 (Xcm3117 and Xcm3755) to 0.795 (Xcm0497) (Table 1).

BIC values derived from the k-means analyses suggested that five is the most appropriate number of clusters (Fig 2; S1 Fig). All individuals had an assignation probability to a given cluster > 0.99. The different genetic clusters were separated from each other by 3–5 locus-variations (Fig 3). The two major DAPC clusters (clusters 3 and 4) grouped strains from many geographic origins (Pacific, South-West Indian Ocean). Strains identified in cluster 1 were geographically restricted to West Africa, whereas clusters 2 and 5 group strains from Asia. Clusters 2 and 4 contained strains previously assigned by AFLP to the B and CD clusters, respectively. Strains previously assigned to the A cluster by AFLP split into clusters 3 and 5.

## Discussion

MBC is a serious disease threatening both mango and cashew, two major agricultural industries in tropical and subtropical regions. Herein, we report a new MLVA-based resolutive and robust genotyping method targeting 16 TR markers, which can be applied to the global surveillance of *Xcm*. The newly developed MLVA-16 genotyping scheme was only slightly less discriminative than AFLP. Data derived from the two datasets were highly correlated, suggesting a fairly good resolution and

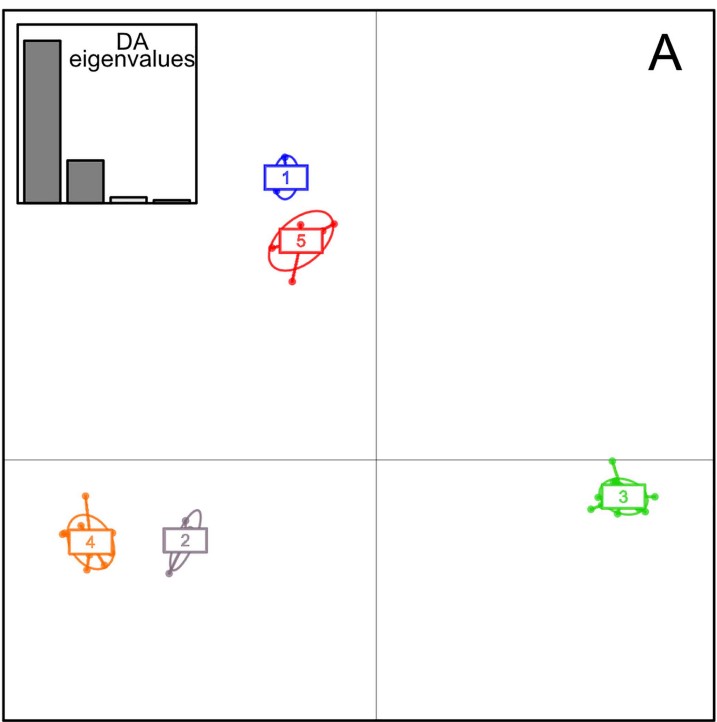

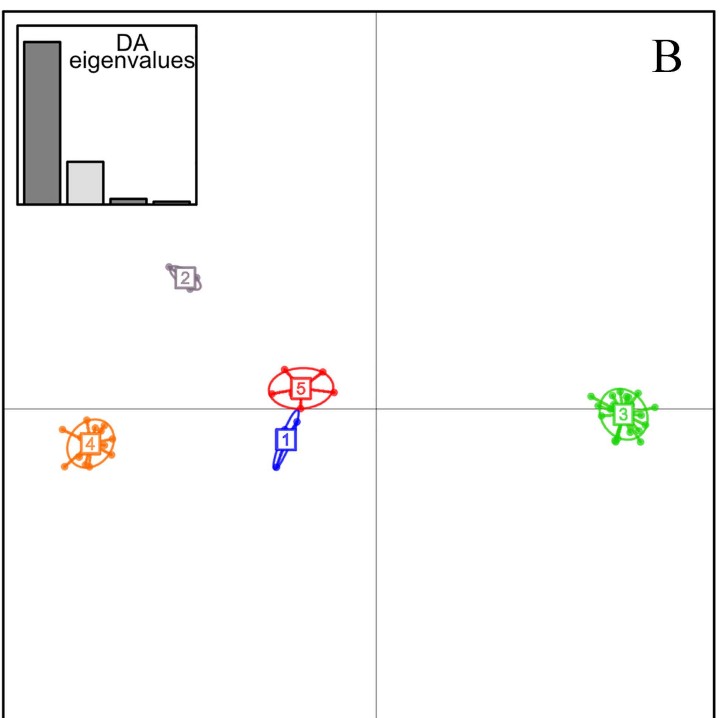

**Fig 2. Genetic structure of a worldwide strain collection of *Xanthomonas citri* pv. *mangiferaeindicae* based on the discriminant analysis of principal components (DAPC) of tandem repeat data.** Numbers and colours represent the five genetic clusters retained from Bayesian information criterion (BIC) values (S1 Fig). A. scatterplot representing axes 1 and 2 of the DAPC; B scatterplot representing axes 1 and 3 of the DAPC.

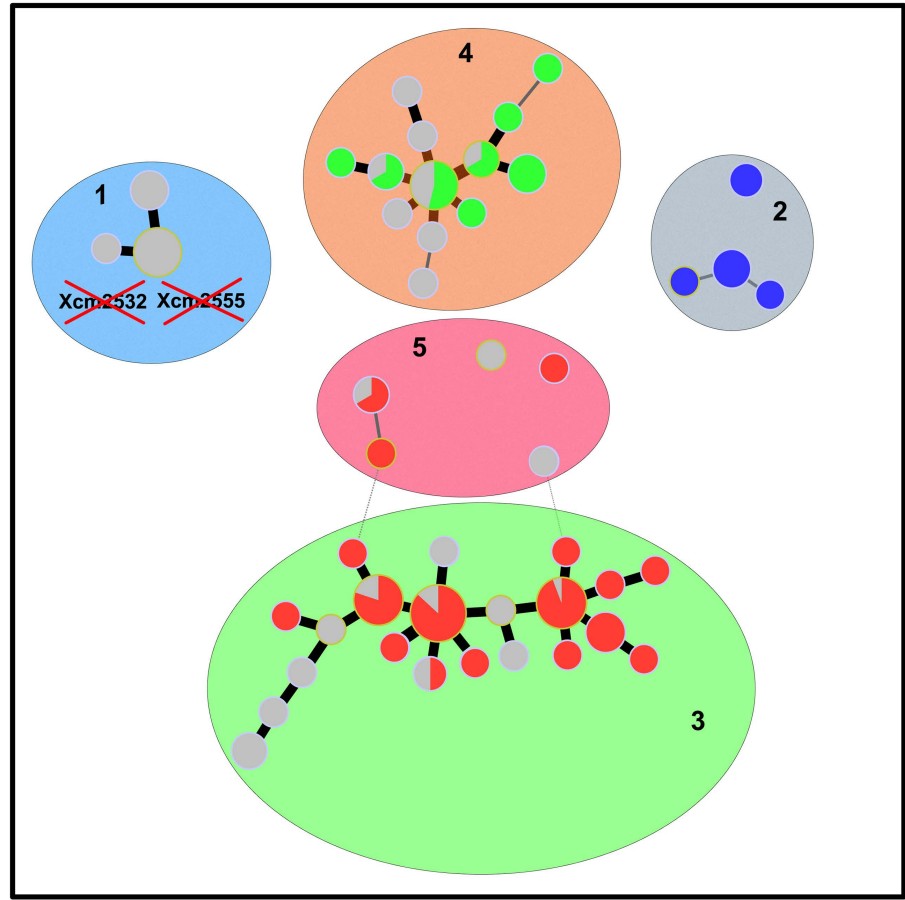

**Fig 3. Minimum spanning tree from MLVA-16 data showing the genetic diversity of a worldwide strain collection of *Xanthomonas citri* pv. *mangiferaeindicae*.** All strains from distinct networks or singletons differed by ≥ 4 TR loci. Dots represent haplotypes. Dot diameter and colour represent the number of strains per haplotype, country and AFLP clusters, respectively (red: A cluster; blue: B cluster; green: CD cluster; grey: AFLP data not available). Lines linking dots represent the amount of polymorphism among haplotypes (thick line = single-locus variation; thin line = double-locus variation; dashed line = triple-locus variation). Coloured background ellipses and numbers represent DAPC clusters, as shown in Fig 2.

phylogenetic signal. Moreover, it allowed us to clearly distinguish *Xcm* from *Xca*, another pathovar pathogenic to Anacardiaceae in the same bacterial species. MLVA-based genotyping has many advantages over AFLP. It is cost-effective, easy to use, portable and provides a satisfactory degree of resolution. MLVA-based genotyping was proven to be useful for the surveillance of xanthomonads [24]. It represents a cheap frontline analysis in situations where massive whole genome sequencing (WGS) cannot be implemented, a common occurrence in countries where mango and cashew industries are economically important. MLVA can also be applied to the selection of strains for WGS. WGS provides a more robust view of the genetic structure of bacterial pathogens. It also allows access to the accessory genome content, including mobile genetic elements. These are major drivers of bacterial adaptation to selection pressures [41–45] and, therefore, are useful for surveillance.

The data produced herein were made publicly available in the MLVABank database (https://webphim.ird.fr/MLVA_Bank/Genotyping/index.php). This should facilitate comparative analyses of outbreak strains, which can then be considered in the context of the global diversity reported for *Xcm* [30,46]. Collectively, MLVA-16 and the previously reported MLVA-12 targeting microsatellites [21] have the ability to produce complementary datasets to match the evolutionary scale and to shed light on the epidemiological question under investigation [6,47,48].

Strains that originated from Anacardiaceae species other than mango clustered in two genetic groups. Based on the DAPC analysis, all strains isolated from Brazilian pepper grouped in cluster 4, which is totally in agreement with earlier AFLP data [22]. Herein, we report the first genetic characterization of strains originating from a related Anacardiaceae species, the large-leaved rhus (*Searsia longipes*), which were also assigned to cluster 4. All strains isolated from cashew grouped in cluster 1. The low diversity of these West African strains, as previously highlighted by microsatellite data [14], suggests that all strains isolated from this host species or mango in this region would cluster in this group. Globally, we revealed that strains grouping in all five delineated clusters were able to cause outbreaks on mango. However, we do not yet know whether they could cause outbreaks on alternative host species. Indeed, strains from mango and Brazilian pepper exhibit some host specificity. The significance of the latter species to act as an efficient inoculum source for mango has not been established [8]. A more thorough understanding of plant-*Xcm* interactions is required to address this question.

MBC causes major disease outbreaks on mango and cashew, with severe consequences for crop yield and quality in many countries, which directly impacts the industries. Yet, it is clearly a neglected disease, as shown by the limited number of published papers on the subject and the lack of public microbial and genomic resources. No available bacterial strains or genomes could be identified that closely matched the strain cluster that had emerged on mango and cashew in West Africa [10]. Therefore, proposing a genetically-sound hypothesis to putatively identify the source of this emergence was precluded. Nevertheless, our study allowed us to reject an early hypothesis suggesting that *Xcm* strains that are now established in West Africa may have originated from South Africa as a result of the long-distance movement of mango propagative material (budwood). Indeed, a deeper analysis of *Xcm* global molecular epidemiology involving new resources is needed to improve our understanding of the pathogen's large-scale expansion. Despite the first report of the disease in South Africa at the beginning of the 20th century, *Xcm* may have originated from Asia, the area of origin of mango [49]. This is suggested by (i) herbarium samples collected from India in the second half of the 19th century, which show typical leaf lesions [11], and (ii) recent phylogenomic analyses [50]. The case study of West Africa, where MBC was very likely absent until the early 2000s and now represents a major constraint for the mango and cashew industries, emphasizes the importance of (i) efficient pre-entry control measures, and (ii) timely genotyping-based surveillance of crop pathogens. We advocate the importance of mobilizing research efforts to improve our understanding of this major pathosystem, which causes severe damage to two cash crops of agricultural importance.

## Supporting information

**S1 Fig. Bayesian information criterion derived from the DAPC k-means analysis performed on *Xanthomonas citri* pv. *mangiferaeindicae* tandem repeat dataset.**
(PDF)

**S1 Table. Bacterial strains used in this study.**
(XLSX)

**S2 Table. Information on the tandem repeat loci selected for subtyping *Xanthomonas citri* pv. *mangiferaeindicae.***
(XLSX)

**S3 Table. Repeat and spacer sequences of CRISPR-like elements detected by CRISPRCasFinder in high-quality *Xanthomonas citri* pv. *mangiferaeindicae* genomes.**
(XLSX)

## Acknowledgments

We would like to thank A. Dereeper and R. Koebnik for their kind support (MLVAbank database). The authors greatly acknowledge the Plant Protection Platform (3P, IBiSA).

## Author contributions

**Conceptualization:** Olivier Pruvost.

**Data curation:** Karine Boyer, Olivier Pruvost.

**Formal analysis:** Karine Boyer, Cyrille Zombre, Laure Payan, Olivier Pruvost.

**Funding acquisition:** Issa Wonni, Olivier Pruvost.

**Investigation:** Karine Boyer, Cyrille Zombre, Laure Payan.

**Methodology:** Karine Boyer, Laure Payan.

**Project administration:** Issa Wonni, Olivier Pruvost.

**Resources:** Karine Boyer, Olivier Pruvost.

**Supervision:** Issa Wonni, Olivier Pruvost.

**Validation:** Cyrille Zombre, Laure Payan, Issa Wonni, Olivier Pruvost.

**Visualization:** Olivier Pruvost.

**Writing – original draft:** Karine Boyer, Olivier Pruvost.

**Writing – review & editing:** Karine Boyer, Cyrille Zombre, Laure Payan, Issa Wonni, Olivier Pruvost.

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
