## [Decision Letter · Decision Letter 0]

13 Aug 2025

*Xanthomonas citri* pv. *mangiferaeindicae*

Dear Dr. Pruvost,

Thank you for submitting your manuscript to PLOS ONE. After careful consideration, we feel that it has merit but does not fully meet PLOS ONE’s publication criteria as it currently stands. Therefore, we invite you to submit a revised version of the manuscript that addresses the points raised during the review process.

We look forward to receiving your revised manuscript.

Kind regards,

Baochuan Lin, Ph.D.

Academic Editor

PLOS ONE

Journal Requirements:

2. Please expand the acronym “ERDF” (as indicated in your financial disclosure) so that it states the name of your funders in full.

We would like to thank A. Dereeper and R. Koebnik for their kind support (MLVAbank database). The European Union (ERDF contract GURDT I2016‐1731‐0006632 and DPP SantéBiodiv project), Conseil Régional de La Réunion and CIRAD provided financial support. The authors greatly acknowledge the Plant Protection Platform (3P, IBiSA).

KB and OP =>

The European Union (ERDF contract GURDT I2016‐1731‐0006632 and DPP SantéBiodiv project) https://commission.europa.eu/funding-tenders/find-funding/eu-funding-programmes/european-regional-development-fund-erdf_en

Conseil Régional de La Réunion https://regionreunion.com

Centre de coopération internationale en recherche agronomique pour le développement (CIRAD) https://www.cirad.fr

I certify that the sponsors or funders play any role in the study design, data collection and analysis, decision to publish, or preparation of the manuscript

Reviewers' comments:

Reviewer's Responses to Questions

**Comments to the Author**

1. Is the manuscript technically sound, and do the data support the conclusions?

Reviewer #1: Yes

Reviewer #2: Yes

2. Has the statistical analysis been performed appropriately and rigorously?

Reviewer #1: Yes

Reviewer #2: Yes

3. Have the authors made all data underlying the findings in their manuscript fully available?

Reviewer #1: Yes

Reviewer #2: Yes

4. Is the manuscript presented in an intelligible fashion and written in standard English?

Reviewer #1: Yes

Reviewer #2: Yes

Reviewer #1: Major comments

*Xanthomonas citri* pv. *mangiferaeindicae* (Xcm) is the causal agent of bacterial canker, also named bacterial black spot or angular spot, in some species of Anacardiaceae family. With the worldwide introduction and expanded cultivation of certain Anacardiaceae plants with significant economic value, bacterial canker disease has also been continuously spreading and worsening. However, compared to many other xanthomonad pathogens, Xcm has not received sufficient attention. Molecular typing is the basis for population genetics and tracing of the pathogens. The genetic structure of Xcm was primarily dissected by amplified fragment length polymorphism (AFLP), an old typing method with randomness and lacking of sequence information. The development of genomics has provided some new typing methods with more precise, faster, higher in resolution, and easier to operate. In this study, Boyer and her colleagues developed a new tandem repeat-based genotyping scheme targeting 16 TR markers in Xcm, and the MLVA-16 dataset was deposited in a dedicated online database. The MLVA-16 genotyping scheme was only slightly less discriminative than AFLP. Data derived from the two datasets were highly correlated, suggesting a fairly good resolution and phylogenetic signal. This study will further expand our understanding of the genetic diversity of Xcm, and will be helpful for disease diagnosis, early warning, and quarantine of this neglected bacterial pathogen with agricultural risks. The manuscript is well designed and written, and the content is suitable for publication in PLoS ONE. However, there are still some parts that are worth polishing and improving.

Questions and suggestions:

1.Tandem repeats are found throughout the genomes, in which the number of repeat units can vary due to slipped-strand mispairing during DNA replication, leading to length polymorphisms. TR typing is manifesting as one kind of genetic fingerprinting of individual organism, indicating that the individual has specific TR markers. It is suggested to construct a pan-MLVA marker library (MLVAbank database) using several representative and high-quality sequenced Xcm strains other than one single Xcm strain. This additional task will not affect the original MLVA-16 genotyping scheme in the manuscript.

2.The manuscript is about the establishing an experimental method or scheme? which needs to have a clear logic and a well-defined technical route. It is recommended to add a technical route map or a research scheme map in the first part of Results.

3.There are only one table and two figures in the main text of this manuscript, seeming somewhat thin and lacking in information. It is better to add a genomic collinear diagram of several representative strains, along with each TR marker in certain genome.

4.Please also pay attentions to the typo errors, including misplaced space and punctuation marks, e.g., Line 30, misplaced space; Line 279, [24]]; Line 148, 2X, 2×; Line 149, 5X, 5×; Line 170, 10X, 10×.

Reviewer #2: Thank you for submitting an interesting and useful study. This research highlights the relevance of using MVLA as a tool to study the population diversity of bacterial pathogens.

I have some really minor comments:

lines 69 and 70 Vietnam not Viet Nam

line 78 should be Gram not gram as it is named affer a person

Line 79 incomplete sentence, unlike most Xanthomonas species?

Line 297 what Rhus species?

**Do you want your identity to be public for this peer review?** For information about this choice, including consent withdrawal, please see our Privacy Policy

Reviewer #1: **Yes: ** Dr. Yong-Qiang He

Reviewer #2: **Yes: ** Teresa Coutinho

---

## [Author Response · Author response to Decision Letter 1]

27 Sep 2025

Reviewer #1: Major comments

*Xanthomonas citri* pv. *mangiferaeindicae* (Xcm) is the causal agent of bacterial canker, also named bacterial black spot or angular spot, in some species of Anacardiaceae family. With the worldwide introduction and expanded cultivation of certain Anacardiaceae plants with significant economic value, bacterial canker disease has also been continuously spreading and worsening. However, compared to many other xanthomonad pathogens, Xcm has not received sufficient attention. Molecular typing is the basis for population genetics and tracing of the pathogens. The genetic structure of Xcm was primarily dissected by amplified fragment length polymorphism (AFLP), an old typing method with randomness and lacking of sequence information. The development of genomics has provided some new typing methods with more precise, faster, higher in resolution, and easier to operate. In this study, Boyer and her colleagues developed a new tandem repeat-based genotyping scheme targeting 16 TR markers in Xcm, and the MLVA-16 dataset was deposited in a dedicated online database. The MLVA-16 genotyping scheme was only slightly less discriminative than AFLP. Data derived from the two datasets were highly correlated, suggesting a fairly good resolution and phylogenetic signal. This study will further expand our understanding of the genetic diversity of Xcm, and will be helpful for disease diagnosis, early warning, and quarantine of this neglected bacterial pathogen with agricultural risks. The manuscript is well designed and written, and the content is suitable for publication in PLoS ONE. However, there are still some parts that are worth polishing and improving.

We thank reviewer#1 for his/her kind comments.

Questions and suggestions:

1.Tandem repeats are found throughout the genomes, in which the number of repeat units can vary due to slipped-strand mispairing during DNA replication, leading to length polymorphisms. TR typing is manifesting as one kind of genetic fingerprinting of individual organism, indicating that the individual has specific TR markers. It is suggested to construct a pan-MLVA marker library (MLVAbank database) using several representative and high-quality sequenced Xcm strains other than one single Xcm strain. This additional task will not affect the original MLVA-16 genotyping scheme in the manuscript.

Indeed, this was partly achieved in the original submission, as MLVA data from the high-quality genome sequences were already included in the MLVAbank online database. We did not construct a new database in MLVAbank, as suggested, as we believe that it is of limited interest to the readership. Instead, we include in the revised version a table summarizing the total minisatellite content of each sequenced strain for reviewers’ appreciation (including the loci that were not selected for MLVA-16 typing), as determined by Tandem Repeat Finder and the filtering criteria we used. If the editor feels that it is of sufficient interest, it may be included as a supplementary table in the published version of our manuscript.

2.The manuscript is about the establishing an experimental method or scheme? which needs to have a clear logic and a well-defined technical route. It is recommended to add a technical route map or a research scheme map in the first part of Results.

In our opinion, the input of the present study is more providing a new genotyping scheme rather than providing an experimental method, as our analytical procedure was really standard, and is highly similar to the one used previously in many molecular epidemiological studies of human, animal or plant bacterial pathogens. Nevertheless, we followed reviewer #1’s recommendation and included at the beginning of the Results section a short paragraph summarizing our technical route.

3.There are only one table and two figures in the main text of this manuscript, seeming somewhat thin and lacking in information. It is better to add a genomic collinear diagram of several representative strains, along with each TR marker in certain genome.

A new Fig. 1 was produced as requested.

4.Please also pay attentions to the typo errors, including misplaced space and punctuation marks, e.g., Line 30, misplaced space; Line 279, [24]]; Line 148, 2X, 2×; Line 149, 5X, 5×; Line 170, 10X, 10×.

Modified as requested.

Reviewer #2: Thank you for submitting an interesting and useful study. This research highlights the relevance of using MVLA as a tool to study the population diversity of bacterial pathogens.

I have some really minor comments:

lines 69 and 70 Vietnam not Viet Nam

line 78 should be Gram not gram as it is named affer a person

Line 79 incomplete sentence, unlike most Xanthomonas species?

Line 297 what Rhus species?

All editorial suggestions placed by reviewer #2 were included in the revised version. The vernacular name of Searsia longipes remains the large-leaved rhus, although its genus name was modified from Rhus to Searsia in 2007 and further validated in Catalog of Life.

---

## [Decision Letter · Decision Letter 1]

30 Oct 2025

A new tandem repeat-based genotyping scheme for the global surveillance of *Xanthomonas citri* pv. *mangiferaeindicae*, an understudied bacterial pathogen of major importance to mango and cashew production

PONE-D-25-23106R1

Dear Dr. Pruvost,

We’re pleased to inform you that your manuscript has been judged scientifically suitable for publication and will be formally accepted for publication once it meets all outstanding technical requirements.

Kind regards,

Baochuan Lin, Ph.D.

Academic Editor

PLOS ONE

Additional Editor Comments (optional):

Reviewers' comments:

Reviewer's Responses to Questions

**Comments to the Author**

Reviewer #2: All comments have been addressed

2. Is the manuscript technically sound, and do the data support the conclusions?

Reviewer #2: Yes

3. Has the statistical analysis been performed appropriately and rigorously?

Reviewer #2: Yes

4. Have the authors made all data underlying the findings in their manuscript fully available?

Reviewer #2: Yes

5. Is the manuscript presented in an intelligible fashion and written in standard English?

Reviewer #2: Yes

Reviewer #2: Well written and experimental sound manuscript on an understudied bacterial pathogen. Just out of interest, Xcm was the topic of a DSc thesis in the early 1900s by Ethel M Doidge, a South African plant pathologist and the first woman to receive a doctorate in SA. She later went on to become a leading fungal taxonomist.

**Do you want your identity to be public for this peer review?** For information about this choice, including consent withdrawal, please see our Privacy Policy

Reviewer #2: **Yes: ** Teresa Ann Coutinho

---

## [Editor Report · Acceptance letter]

PONE-D-25-23106R1

PLOS ONE

Dear Dr. Pruvost,

I'm pleased to inform you that your manuscript has been deemed suitable for publication in PLOS ONE. Congratulations! Your manuscript is now being handed over to our production team.

Kind regards,

on behalf of

Dr. Baochuan Lin

Academic Editor

PLOS ONE